# Self-management interventions for adults living with Chronic Obstructive Pulmonary Disease (COPD): The development of a Core Outcome Set for COMPAR-EU project

Estela Camus-García[1], Ana Isabel González-González[1,2,3]*, Monique Heijmans[4], Ena Niño de Guzmán[5], Claudia Valli[6,7], Jessica Beltran[7], Hector Pardo-Hernández[8], Lyudmil Ninov[9], Valentina Strammiello[9], Kaisa Immonen[9], Dimitris Mavridis[10], Marta Ballester[1,3], Rosa Suñol[1,3], Carola Orrego[1,3]

1 Avedis Donabedian Research Institute (FAD), Universitat Autonòma de Barcelona, Barcelona, Spain, 2 Institute of General Practice, Goethe University, Frankfurt, Germany, 3 Red de Investigación en Servicios de Salud en Enfermedades Crónicas (REDISSEC), Spain, 4 Netherlands Institute for Health Services Research (NIVEL), Utrecht, The Netherlands, 5 Iberoamerican Cochrane Centre Barcelona, Department of Clinical Epidemiology and Public Health, Biomedical Research Institute San Pau (IIB Sant Pau), Barcelona, Spain, 6 Department of Paediatrics, Obstetrics, Gynaecology and Preventive Medicine, Universidad Atónoma de Barcelona, Barcelona, Spain, 7 Iberoamerican Cochrane Centre Barcelona, Biomedical Research Institute San Pau (IIB Sant Pau), Barcelona, Spain, 8 Iberoamerican Cochrane Centre Barcelona, Biomedical Research Institute San Pau (IIB Sant Pau) - CIBER Epidemiología y Salud Pública (CIBERESP), Barcelona, Spain, 9 European Patients' Forum (EPF), Brussels, Belgium, 10 Department of Primary Education, University of Ioannina, Ioannina, Greece

* aigonzalez@fadq.org

## Abstract

### Background

A large body of evidence suggests that self-management interventions (SMIs) may improve outcomes in chronic obstructive pulmonary disease (COPD). However, accurate comparisons of the relative effectiveness of SMIs are challenging, partly due to heterogeneity of outcomes across trials and uncertainty about the importance of these outcomes for patients. We aimed to develop a core set of patient-relevant outcomes (COS) for SMIs trials to enhance comparability of interventions and ensure person-centred care.

### Methods

We undertook an innovative approach consisting of four interlinked stages: i) Development of an initial catalogue of outcomes from previous EU-funded projects and/or published studies, ii) Scoping review of reviews on patients and caregivers' perspectives to identify outcomes of interest, iii) Two-round Delphi online survey with patients and patient representatives to rate the importance of outcomes, and iv) Face-to-face consensus workshop with patients, patient representatives, health professionals and researchers to develop the COS.

### Results

From an initial list of 79 potential outcomes, 16 were included in the COS plus one supplementary outcome relevant to all participants. These were related to patient and caregiver

**Data Availability Statement:** All relevant data are within the paper and its Supporting information files.

**Funding:** This work was supported by the EU Horizon 2020 research and innovation programme (grant agreement no. 754936).

**Competing interests:** The authors have declared that no competing interests exist.

knowledge/competence, self-efficacy, patient activation, self-monitoring, adherence, smoking cessation, COPD symptoms, physical activity, sleep quality, caregiver quality of life, activities of daily living, coping with the disease, participation and decision-making, emergency room visits/admissions and cost effectiveness.

## Conclusion

The development of the COPD COS for the evaluation of SMIs will increase consistency in the measurement and reporting of outcomes across trials. It will also contribute to more personalized health care and more informed health decisions in clinical practice as patients' preferences regarding COPD outcomes are more systematically included.

## Introduction

Chronic Obstructive Pulmonary Disease (COPD) is one of the major causes of morbidity and mortality worldwide [1,2]. The economic and social burden related to COPD are expected to increase over the coming decades due to the continued exposure to COPD risk factors and the increasing aging of the world's population [3]. COPD prevalence varies across countries and across different groups within countries (i.e., being male, older and former or current smoker) [4]. It is directly related to the prevalence of tobacco smoking, although in many countries outdoor and indoor air pollution constitute major risk factors [5,6].

The literature suggests that self-management interventions (SMIs) may improve clinical outcomes, quality of life and reduce costs of chronic conditions, including COPD [7,8]. A Cochrane systematic review showed that SMIs along with support from health professionals improve health-related quality of life while decreasing hospitalizations and emergency department visits of COPD patients [9].

Two recent studies, the COMET [10] and the PIC-COPD [11] showed the potential of SMIs for reducing exacerbations and mortality in integrated case management, as well as for increasing physical activity. However, synthesizing the evidence on the relative effectiveness of SMIs for COPD is challenging due to heterogeneity of interventions, lack of clear definitions of self-management components, and variability in the outcomes reported. Moreover, systematic reviews on SMI effectiveness have found insufficient data for some outcomes, which may be suggestive of selective reporting [9,12,13].

SMIs can only be compared across studies when they share some common outcomes. In addition, it is important to create consensus about what outcomes are especially relevant to assess the effects of SMI and how they should be measured. By reaching consensus of a standardized set of outcomes that should be minimally measured and reported in future COPD clinical trials, we will ensure the comparativeness of results and synthesis of the evidence across studies [14]. This outcome set should be relevant for all stakeholders, but especially for patients, as they are the ones primarily responsible for the daily management of their disease. In this study we propose a systematic approach to develop a Core Outcome Set [COS] for measuring the effectiveness of SMIs interventions in COPD from the perspective of both patients and health care professionals. This study is part of COMPAR-EU, an EU-funded project designed to bridge the gap between current knowledge and practice on SMIs in four chronic conditions including COPD.

## Material and methods

The COS for SMIs in COPD patients was developed in accordance with the Core Outcome Measures for Effectiveness Trials (COMET) Handbook [14] and the Core Outcome Set-STAndards for Development (COS-STAD) guidelines [15]. This study was conducted according to a protocol previously published [16]. The COMPAR-EU COS approach involved four inter-linked stages that are described below and summarized in Fig 1.

### Stage 1. Development of an initial catalogue of outcomes

**Data sources and searches.** We developed an initial catalogue of outcomes from a literature review of two overviews of systematic reviews evaluating the effectiveness of SMIs for

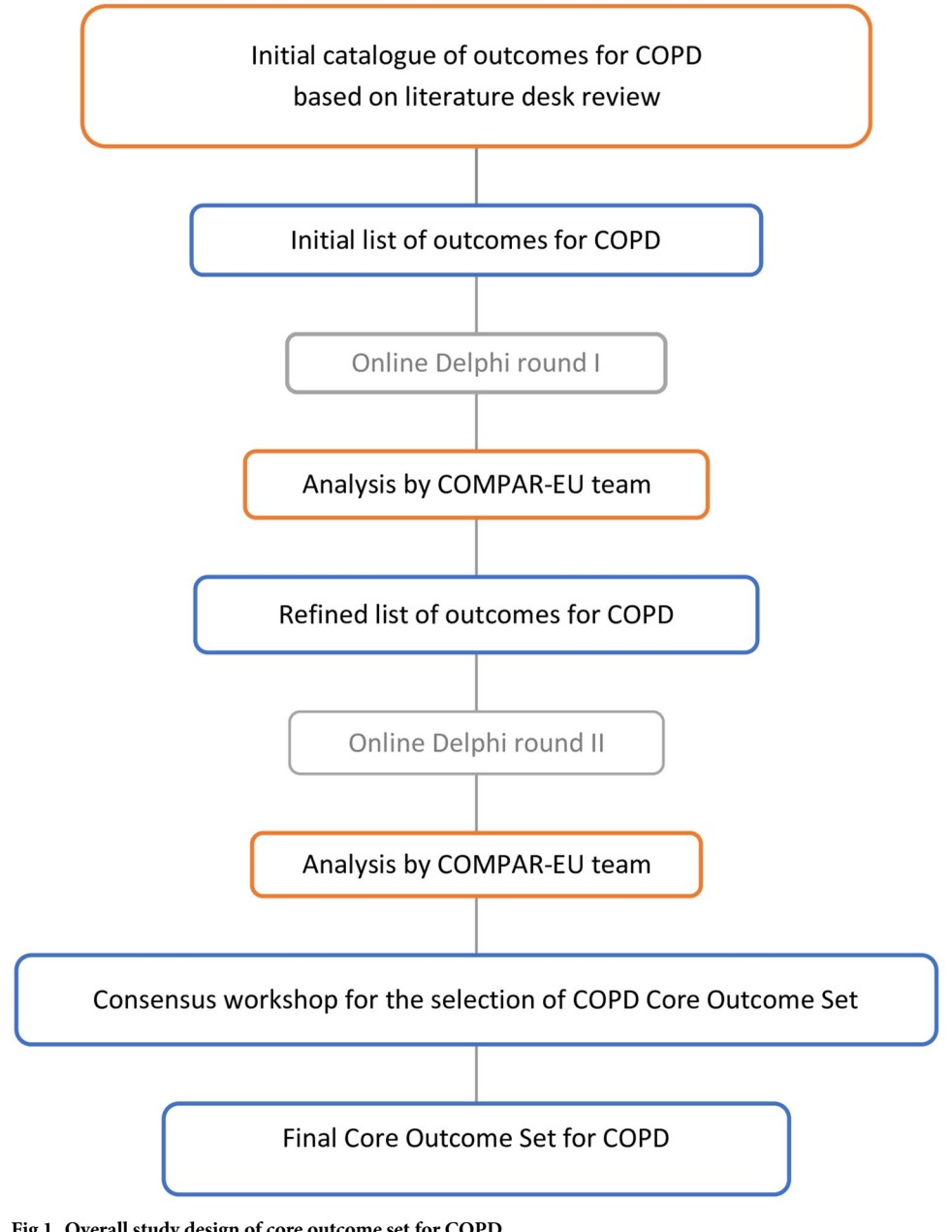

**Fig 1. Overall study design of core outcome set for COPD.**

chronic diseases: i) PRO-STEP (Promoting Self-Management for Chronic Diseases in the EU) [17] and ii) EMPATHiE (Empowering Patients in the Management of Chronic Diseases) [18]. Both reviews [17,18] were performed by the research team and were considered as the starting data source to build the initial list of outcomes. We additionally searched for COPD COS in relevant organizations databases such as COMET [10] and ICHOM [International Consortium for Outcomes Health Measurement] [19], to discard the existence of COS on this area and avoid work duplication as recommended by the COMET handbook [20]. The syntax used for the additional literature review in PubMed was the following: (pulmonary disease, chronic obstructive "[MeSH Terms] AND "patient preference"[MeSH Terms]) AND "outcome assessment (health care)"[MeSH Terms]; "pulmonary disease, chronic obstructive"[MeSH Terms] AND "core outcome set"[All Fields].

**Study selection.** We included systematic reviews and individual studies that reported outcomes on SMIs for patients with COPD. We excluded systematic reviews that did not report a final list of outcomes or individual studies where the final list of outcomes was not developed considering patients' input, experiences or values and preferences.

We screened title and abstracts and assessed eligible full-text articles independently. In case of disagreement, reviewers reached consensus or consulted with a third reviewer from the review team. Reviewers checked references from included studies to identify other potentially eligible studies.

**Data extraction.** Pairs of authors independently extracted the following data from eligible studies: i) study database, ii) type of publication (i.e., published COS, literature review or systematic review), iii) age groups, and iv) list of outcomes.

**Data synthesis.** We tabulated and classified the identified outcomes into the following seven categories following the process for the development of the COMPAR-EU taxonomy [21]: i) empowerment components, ii) adherence to expected self-management behaviours, iii) clinical outcomes, iv) patient and informal caregivers' quality of life, v) perceptions and/or satisfaction with care, vi) healthcare use and vii) costs. The research team reviewed and discussed outcomes and merged them when possible.

Through an iterative process, an external clinician and researcher reviewed and discussed the resulting list of outcomes with multidisciplinary experts from the COMPAR-EU consortium. We prepared a definition of each outcome with the participation of all COMPAR-EU team members. Experts in health literacy and patient representatives adapted the resulting list of outcomes and presented it in plain language. This list of outcomes was to be used in the first round of the Delphi process (Stage 3).

## Stage 2. Scoping review of reviews on perspectives of patients and their caregivers regarding self-management

We conducted a scoping review of reviews [22] to identify and describe key concepts related to outcomes by exploring patients' and caregivers' preferences and experiences when coping with COPD and its self-management.

**Data sources and searches.** We searched MEDLINE, CINAHL and PsycINFO from inception until February 2018. We applied a content search strategy for values and preferences [23] in combination with terms specific for COPD. We used review filters available in each database. We included the following terms for identifying patients' perspectives: patient perception, experience, perspective, understanding, preferences and health utilities.

**Study selection.** We included reviews of quantitative, qualitative or mixed-methods studies that explored the perspectives, experiences, values and preferences of patients and caregivers on SMIs for COPD.

**Data extraction.**   In a previously pilot-tested data extraction form, we collected the general characteristics and main findings of each review.

**Data synthesis.**   We conducted a descriptive thematic synthesis including the identification of codes, descriptive themes and main themes relevant to outcomes of SMI for COPD. We paired main emerging themes with the subdomains of the COMPAR-EU taxonomy [21] and mapped the correspondence between themes and the initial catalogue of outcomes. We developed infographics illustrating themes to be used as aid materials during the consensus workshop.

## Stage 3. Delphi survey (Round I and II)

To prioritize the outcomes identified, we administered two-round modified Delphi online surveys to a convenience sample. Our sample included patients and patient representatives to ensure that we address outcomes that matter to patients as well as to other stakeholders.

**Study population and eligibility criteria.**   We included adults diagnosed with COPD and patient representatives who were able to understand and speak English and provided informed consent to participate through the web platform hosting the Delphi rounds. We made efforts to recruit patients considering age, gender, geographical location and education. However, the patients who participated in this study may have been more knowledgeable, motivated and aware of treatment options and legislation than other COPD patients. In the other hand, they may have been more motivated to engage in research and advocacy activities. They may have also been more aware of the needs of other COPD patients and during the discussion it was evident that they wanted to represent the views of COPD patients as a whole and not just their own. As an example, they mentioned that while they were aware of strategies to avoid exacerbations, other patients may be less knowledgeable.

**Recruitment strategy.**   Participants were identified within the European Patients' Forum's EU wide membership network of more than 70 patient organizations [24] and other patient groups (e.g., those involved in ICHOM) [19]. Recruitment started and concluded in February 2018 and ended in May 2018.

**Delphi survey.**   The first and second Delphi rounds took place between May 2018 and June 2018. All participants received an online survey with the outcomes and definitions. They also received weekly reminders and were able to return to the questionnaire within a 3-week period. Some of the participants were supported by their local organizations when completing it. Participants were asked "*How important do you think the following outcomes are to measure the success of self-management in people with COPD*?". COPD outcomes for SMIs were prioritized during the two-round Delphi process using a 1 to 9 Likert scale (1 being the least and 9 being the most important for the self-management of COPD).

During the second round, participants were able to see ratings (average score) from the first round and thus, adjust, confirm or rethink their answers. They were also allowed to deliberate. This process enabled participants to rate the most relevant SMIs outcomes for COPD according to their perspective.

**Data synthesis and analysis.**   All outcomes were categorized into three groups based on the level of agreement of ratings from the two-round Delphi online surveys as follows (Table 1): i) Group 1 or "high consensus and high importance outcomes", ii) Group 2 or "low consensus and mixed importance outcomes" and iii) Group 3 or "high consensus of moderate and low importance outcomes". We used 70% as a cut off for high agreements based on GRADE recommendations, COMET guidelines and previous papers reporting patient-centred core outcome sets that also used these thresholds [20,25,26].

**Table 1. Categories of outcomes by level of agreement.**

| Group | Votes | | Interpretation |
|---|---|---|---|
| Group 1a | ≥ 70% voted 8–9 | ≤ 15% voted 1–3 | High agreement on high importance. |
| | | | Suggestion to include on Core Outcome Set |
| Group 1b | ≥ 70% voted 7 | ≤ 15% voted 1–3 | High agreement on high importance. |
| | | | Suggestion to include on Core Outcome Set |
| Group 2 | Intermediate results | | Inclusion or exclusion on Core Outcome Set to be decided in consensus workshop |
| Group 3 | ≤ 15% voted 8–9 | ≥ 70% voted 1–3 | High agreement on moderate or low importance. |
| | | | Suggestion to exclude from Core Outcome Set |

## Stage 4. Consensus workshop and final COPD COS

The final stage of the COPD COS development was a two and a half-day, in-person consensus workshop held in July 2018 in Berlin (Germany). The aim of the workshop was achieving consensus on the most important outcomes to include in the final COPD COS for the COMPAR-EU project. COPD patients and patient representatives who participated in the two Delphi rounds, health professionals and researchers were invited to participate. Researchers and health professionals were selected from a purposive sample of a heterogeneous group of health professionals representing relevant specialties on the care of patients with COPD (general practitioners, specialists, nurses...) and researchers that came from seven collaborating partner-teams, who knew the process well and could participate on the ultimate objective of facilitating dialogue between patients, patient representatives and health professionals during the consensus workshop.

Participants received the results of the two-round Delphi survey (stage 3), and infographics illustrating themes, by outcome, from the scoping review (stage 2) one week before the consensus workshop which were used as additional material for free consultation. We organised outcomes according to a preliminary version of the outcome COMPAR-EU taxonomy [21]. We sorted them by level of agreement as described previously. The COMPAR-EU research team led step-by-step the flow of the discussion to address potential discrepancies across stakeholders (Fig 2). The group worked on prioritizing and selecting a maximum of 15 outcomes and up to five supplementary outcomes from those that had remained from the Delphi survey results. Participants selected outcomes through an iterative voting (secret vote) and discussion process. Outcomes that were closely related were merged. Once a preliminary list was agreed upon after voting and discussing, participants further reviewed the included outcomes and reached an agreement on the final version of the COPD COS.

**Ethics statement.** Ethical approval was obtained by the Clinical Research Ethics Committee of University Institute for Primary Care Research–IDIAP Jordi Gol in March 2018. All patients and other stakeholders provided written informed consent prior to participation.

## Results

### Stage 1. Development of an initial catalogue of outcomes

**Study selection.** The literature review of previous EU funded projects (PRO-STEP [17] and EMPATHiE [18]) identified records focusing on SMIs in chronic diseases in general. We included 22 systematic reviews specific to COPD [27–48] from PRO-STEP. The additional search in COMET [10], ICHOM [19] and snowballing, which included i) looking at suggestions of similar studies in the search databases, ii) looking at the references of eligible studies, and iii) re-running searches using terms from eligible studies, yielded 23 articles. After full-text

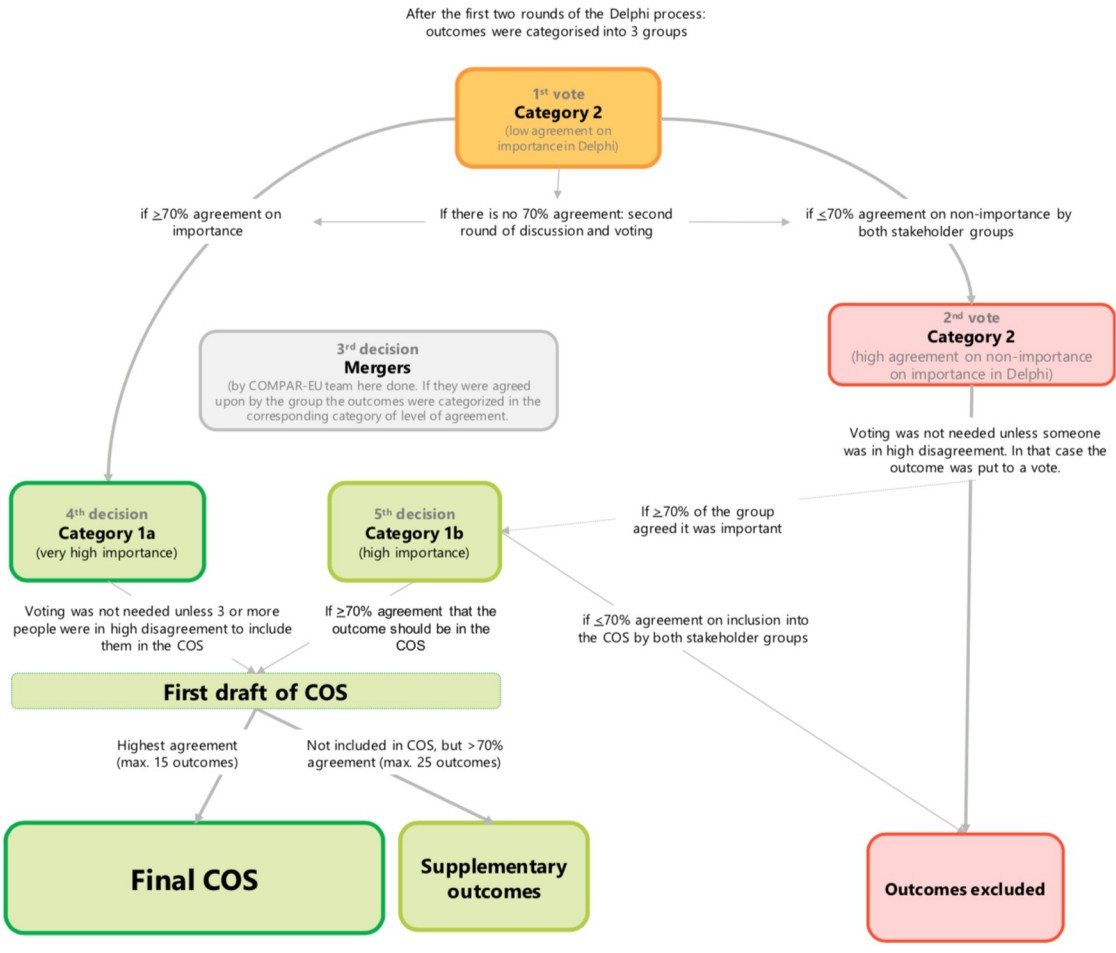

**Fig 2. Consensus workshop decision manual.**

appraisal, we included five studies [49–53]; one study was excluded because it did not report the list of outcomes [54].

**Study characteristics.** The five included studies reported: i) a summary of outcomes for COPD pharmacological trials from lung function to biomarkers created by the American Thoracic Society/European Respiratory Society Task Force [49], ii) a review of instruments used to measure symptom response in pharmacological trials [50], iii) a review of articles determining themes identified as most important by COPD patients for any aspect of care of COPD [51], iv) a review assessing clinical outcomes in COPD mainly used on current published data [53], and v) a study addressing patient preferences regarding the expectations related to treatment of COPD.

**List of outcomes and outcomes classification.** We identified 79 outcomes for the initial list of outcomes. We classified outcomes into seven predefined subdomains based on a taxonomy for SMIs [21]. Table 2 presents the outcomes classification.

## Stage 2. Scoping review of reviews on perspectives of patients and their caregivers regarding self-management

**Study selection.** Among the 1,031 unique screened references, 27 reviews were included comprising more than 800 studies.

**Table 2. List of COPD outcomes and classification.**

| Subdomain | Outcome |
| --- | --- |
| Basic empowerment components | 1  Patient activation |
| | 2  Self-efficacy |
| | 3  Knowledge |
| | 4  Health literacy |
| | 5  Caregiver knowledge |
| | 6  Caregiver self-efficacy |
| Level of adherence to expected self-management behaviors | 7  Taking medication or other treatment as advised (adherence) |
| | 8  Self-monitoring |
| | 9  Diet habits |
| | 10 Diet habits (adherence to diet) |
| | 11 Physical activity |
| | 12 Smoking cessation |
| | 13 Smoking |
| Clinical outcomes | 14 Body weight |
| | 15 Malnutrition |
| | 16 Tiredness (fatigue) |
| | 17 Interrupted |
| | 18 Sleep problems sleep (disturbed sleep) |
| | 19 Sleep quality |
| | 20 Sleepiness |
| | 21 Chest tightness or discomfort |
| | 22 COPD symptoms (short term) |
| | 23 COPD symptoms (long term) |
| | 24 Breathlessness (Dyspnea) |
| | 25 Exacerbation |
| | 26 Lung function (FEV1, FVC) |
| | 27 Lung function |
| | 28 Lung function (LTOT) |
| | 29 Lung function/CPAP |
| | 30 Muscle strength |
| | 31 Effort test/Exercise capacity |
| | 32 Complications |
| | 33 Treatment side effects (adverse effects) |
| | 34 Mortality |
| Patient and informal caregivers' quality of life | 35 Usual activities |
| | 36 Mobility |
| | 37 Work |
| | 38 Physical activities |
| | 39 Sex life |
| | 40 Normality |
| | 41 Pain or discomfort |
| | 42 Treatment burden |
| | 43 Medication burden |
| | 44 Positive attitude |
| | 45 Depression |
| | 46 Anxiety |
| | 47 Stress |
| | 48 Coping |
| | 49 Hostility |
| | 50 Happiness |

(*Continued*)

**Table 2.** (Continued)

| Subdomain | Outcome |
|---|---|
| | 51 Participation in social activities |
| | 52 Self-esteem |
| | 53 Family relationships |
| | 54 Friends |
| | 55 Social activities |
| | 56 Caregiver quality of life |
| | 57 Caregiver burden |
| | 58 Caregiver anxiety and/or depression |
| Perception of and/or satisfaction with care | 59 Satisfaction with/perception of care |
| | 60 Participation and decision-making |
| | 61 Patient-health care provider relation |
| | 62 Communication with health care professionals |
| | 63 Extent to which the health care professional gives enough time to listen to the patient |
| | 64 The patient feels s/he has enough information |
| Healthcare use | 65 Number of primary care or outpatient (ambulatory) visits |
| | 66 Number of nurse visits |
| | 67 Number of virtual visits or contacts with healthcare providers |
| | 68 Number of visits to specialist doctors |
| | 69 Number of home care visits |
| | 70 Number of visits with other healthcare professionals |
| | 71 (Number of) emergency department visits (hospital) |
| | 72 Number of hospital admissions |
| | 73 The length of time spent in hospital (length of hospital stays) |
| | 74 Number of re-hospitalizations unexpected return to hospital |
| Cost | 75 Impact of healthcare costs for the healthcare system |
| | 76 Cost of hospitalizations for the healthcare system |
| | 77 Cost savings for the healthcare system |
| | 78 Direct medical costs for patient |
| | 79 Value for money of the self-management intervention |

**Study characteristics.** Of the 27 reviews for COPD, 16 (59%) were qualitative evidence synthesis [55–70], six (22%) quantitative systematic reviews [51,71–75], four (15%) were mixed methods research synthesis [76–79], and one (4%) was a literature review [80].

The number of included studies ranged from five [73] to 213 [75]. The majority of the reviews (n = 22, 82%) included only the patients' perspective. The phenomena of interest addressed among reviews were preferences on health states of COPD (n = 5, 18%), experiences with the process of self-management (n = 14, 52%) and experiences with self-management interventions (n = 8, 30%).

**Main themes related to SMI outcomes for COPD.** We identified 21 main themes, which are presented in Table 3. These themes were classified under i) empowerment components, ii) adherence to the expected self-management behaviours, iii) clinical-related outcomes, iv) quality of life of patients and caregivers, v) perceptions and/or satisfaction with care, vi) health care use, vii) costs. Table 3 presents the subdomains of the COMPAR-EU taxonomy and the related identified themes for COPD.

**Table 3. Main themes related to COPD outcomes according to the subdomains from COMPAR-EU taxonomy.**

| Subdomains from the COMPAR-EU taxonomy | Main themes for COPD | References |
|---|---|---|
| Empowerment components | Health knowledge [52,56,57,60,63,64] | [55,58,59,65,66,69] |
| | Help/health-seeking behavior | [55,59,60,69,81] |
| | Technological (digital) literacy | [67] |
| Adherence to expected self-management behaviors | Adherence to treatment | [60] |
| | Self-care ability | [63,77,81,82] |
| | Smoking behavior | [59,60,62,65,74] |
| | Perceived benefit (importance) of the intervention | [57,62,64–67,76,78,79] |
| Clinical-related outcomes | Adverse events | [75] |
| | Mortality | [65] |
| | Progression of the disease | [58–60,71–73,75,81,83] |
| Quality of life of patients and caregivers | Informal caregiver's burden | [59,60,80] |
| | Physical functioning | [57–59,65,69,78,79] |
| | Psychological distress | [58,60,65,67,70,74,77,78,82] |
| | Social support | [55,57,58,62,64–66,70,76,78,79,81,82] |
| Perceptions and/or satisfaction with care | Individualized care | [60] |
| | Patient-provider interaction | [55,58–60,81] |
| | Perceived quality of care | [67,82] |
| | Usability | [62,67] |
| Healthcare use | Access to healthcare | [55,59,69] |
| | Visits or contacts with healthcare professionals | [62,67,82] |
| Costs | Cost for patients (out of pocket) | [75] |

COPD = Chronic Obstructive Pulmonary Disease.

## Mapping of themes

Of the 79 outcomes from the initial catalogue of outcomes, 45 were covered in the thematic synthesis (57%). All outcomes of the subdomain "empowerment components" were informed by the scoping review findings (n = 4, 100%), while the subdomain "costs" was the least informed subdomain (n = 1, 20%). Fig 3 reports the number of outcomes informed by the thematic synthesis of the scoping review.

## COPD infographic

An infographic was developed for the final consensus workshop including the main findings and topic related images (Stage 4). The infographic included the outcomes of the initial catalogue informed by the scoping review, classified according to the preliminary version of the outcome taxonomy (S1 File). This material and results from the Delphi rounds were sent to the consensus workshop participants (stage 4) one week in advance.

## Stage 3. Two-round modified Delphi survey

Participants were invited via email. Nine participants accepted the invitation to participate and completed round I and round II of the Delphi online survey. Of these, five (56%) were patients and four (44%) were patient advocates or patients' representatives. Six (67%) were men, five (56%) were over 65 years old and seven (78%) had higher education (master or doctoral equivalent) (S2 File).

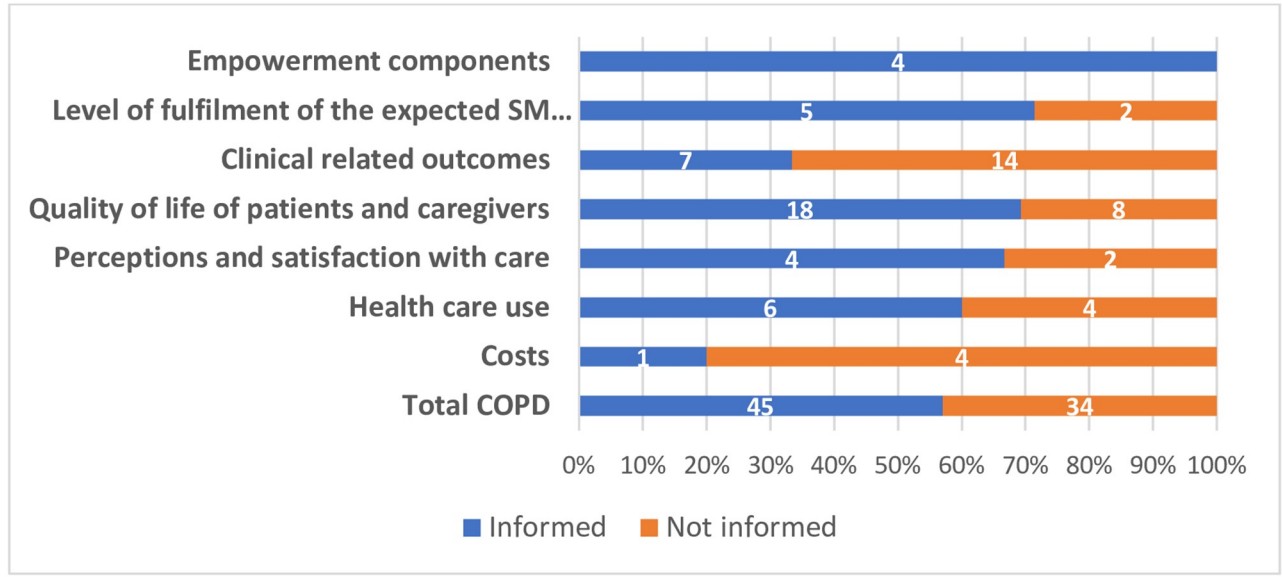

**Fig 3. Mapping of themes per COMPAR-EU taxonomy subdomain.**

After the two-round Delphi survey, 23 (29%) of the 79 included outcomes were voted as high agreement on high importance (Group A: ≥70% of participants voted 7 to 9 on the Likert scale), eight (10%) as high agreement on non-importance (Group C: ≥70% of participants voted 1 to 3 on the Likert scale) and 48 (61%) voted intermediate agreement on importance (Group B) (Table 1).

## Stage 4. Consensus workshop and final COPD core outcome set

Five of the nine patients or patients' representatives that participated in the Delphi online survey and five health professionals and researchers participated in the face-to-face consensus meeting. Five members of the COMPAR-EU research team participated as facilitators (S3 and S4 Files).

The consensus workshop resulted in 16 outcomes for COPD plus 1 supplementary outcome (Table 4). Within these 16 outcomes, Delphi participants rated eight (50%) as high agreement on high importance, seven (44%) as low agreement and mixed importance rating, and one (6%) as high consensus of moderate and low support. Knowledge was part of the high consensus and high importance outcomes and was rendered as a supplementary outcome.

## Discussion

### Main findings

The final COS for COPD included 16 outcomes plus 1 supplementary outcome. It represents the first COS developed based on patient preferences for evaluating SMIs in adults living with COPD. The COS incorporated results from a literature review complemented by a participatory process involving patients and patient representatives along with health professionals and researchers in all stages of the process.

**Table 4. COMPAR-EU COS for COPD.**

| Outcome (COS) | Definition |
|---|---|
| Knowledge (supplementary) | Relates to knowledge about COPD in general and COPD self-management, or the way care for COPD is organized and this both for patients and their social network. |
| Caregiver knowledge and competence | That the caregiver has competences and knowledge of the disease and its management. |
| Self-efficacy | A person's belief that s/he is capable of doing something, often related to a specific goal s/he wants to achieve; feeling of confidence and of being in control. |
| Patient activation | The knowledge, skills and confidence a person has on managing their own health and healthcare, including a feeling of being responsible for taking care of their own health. |
| Self-monitoring | The extent to which a patient (regularly) monitors themselves as agreed with her/his healthcare professionals, for example her/his symptoms or weight. |
| Taking medication or other treatment as advised (adherence) and adherence to regular visits | The extent to which a patient follows the prescribed treatment, such as taking medication as advised and following life-style advice, and extent of attending scheduled visits. |
| Smoking cessation | Stopping smoking (and/or smoking less). |
| COPD symptoms (short term) | Extent of Symptoms relief (in the short-term, including cough; breathlessness, among others). |
| Physical activity—muscle strength | Referral/participation in a Pulmonary Rehabilitation program: Physical activity, Physical activities, Muscle strength linked with exercise capacity plus an overall support. |
| Sleep quality | Sleep quality contains interrupted sleep, sleep problems, sleep quality (as overall) and sleepiness. |
| Exacerbation | Increased breathlessness, mucus/phlegm/sputum production, and change in color of sputum and Feeling out of breath. |
| Caregiver quality of life (including burden) | Caregiver quality of life and the burden that he/she feels from the caregiver's tasks. |
| Activities of daily living: including sex life, social activities and work (usual activities) | Being able to do usual activities, such as personal hygiene, housework, sex, managing finances, social activities and work. |
| Coping with the disease, including depression and anxiety | How well a person feels able to cope/manage with stress or other difficulties caused by the disease, including depression and anxiety. |
| Participation and decision making | Feeling able to participate actively in her/his own care (as much as s/he wishes). |
| Number of emergency room visits and admissions | Number of visits to emergency department visits and hospital admissions. |
| Cost effectiveness and resources use | It includes value for money of the self-management intervention and the use of resources. |

## Our results in the context of previous research

To the best of our knowledge, this is the first COS where a significant part of the work was led by patient representatives' organizations (EPF). Although various approaches have been described to develop COS [84,85], it is still uncertain which are the most appropriate. We chose to follow an iterative mixed-method approach involving different methodologies used in previous studies [86]. The COS we present is novel since it focuses specifically on SMIs for COPD. Previous studies have focused on COPD management or other conditions [87–89]. Spargo et al. [87] developed a COS for trials investigating the long-term management of bronchiectasis combining an overview of systematic reviews and qualitative studies and a Delphi panel that included mostly health professionals who rated the importance of each outcome initially selected. Verburg et al. [88] developed a standard set of outcome domains and proposed

measures for patients with COPD for Dutch primary care physical therapy using a consensus-driven modified RAND-UCLA appropriateness method with relevant stakeholders. Jones et al. [89] created a priority list of measures for a combined COPD and heart failure exercise rehabilitation program through a stakeholders consensus event.

### Strengths and limitations

The first list of COS was mainly based on the results from a literature review on three comprehensive overviews of systematic reviews performed in a previous project (PRO-STEP). As such, it incorporates a robust body of evidence vested in previous projects. The COS development aligns with current methodological guidelines for COS development, as it included a participatory process of patients, patient representatives and other key stakeholders in all stages of the process [14]. Therefore, the resulting COS is strongly based on patient preferences while also incorporating the viewpoints of health professionals, researchers and patients' representatives.

Outcome definitions were adapted to patient accessible language by EPF, which has extensive experience working with and presenting research material to patients in an intelligible manner. This ensured the comprehensibility of the process and the applicability of the results.

Our work is subject to some limitations. The number of participants during the Delphi process was small but the minimum number of patients that had to complete the two Delphi rounds was achieved. We are confident that this shortcoming was overcome via the further deliberations that took place during the workshop. For the workshop, since only five of nine patients from Delphi participated in the consensus, we cannot rule out potential of attrition bias. Lastly, our sample in the Delphi and the consensus workshop may not be entirely representative of the population of patients with COPD. They could represent very motivated individuals or well-informed patients with high education or digital skills. However, and given the resources available, it would not have been feasible to adopt methodology different from electronic surveys (e.g., in-person interviews or surveys) to reach out to participants that are more diverse.

### Implications for practice and research

The identified COS will inform a series of systematic reviews and network meta-analysis (NMA) about the effectiveness of SMIs as part of the COMPAR-EU project. We are confident that the COPD COS reflects the preferences of all key stakeholders and that it might be applicable with context adaptation to wide range of settings across Europe and the world. Future research evaluating SMIs for COPD should, as a minimum, include the outcomes in the proposed COS. Further work is needed to identify and provide guidance on the most appropriate measures for each outcome, on the right instruments or approaches to measure these outcomes, and on the length of follow up. Moreover, it will be important to identify strategies for fostering the collection of this information, the role of the different providers, and the settings where these outcomes can be assessed.

### Conclusions

We have developed the first COS for SMIs in COPD. This COS will increase consistency in the reporting of results that are relevant to patients across trials evaluating SMIs for COPD. This COS will enhance evidence synthesis of COPD patient-relevant outcomes and will decisively support research and overall field development. It will improve informed health-decision making in clinical practice and will increase the certainty of evidence to guide policy-making and clinical practice regarding SMI in COPD patients.

## Supporting information

**S1 File. Infographic COPD.**
(PDF)

**S2 File. Delphi online survey participant characteristics.**
(PDF)

**S3 File. Consensus workshop participants characteristics–patients/patient representatives.**
(PDF)

**S4 File. Consensus workshop participants characteristics–health professionals and researchers.**
(PDF)

## Acknowledgments

The authors would like to thank all patients and patient representatives who participated in this project. Their real life-experiential knowledge was invaluable for the development of this COS.

The COMPAR-EU group: Aretj-Angeliki Veroniki (University of Ioannina, Department of Primary Education), Carlos Canelo-Aybar (Iberoamerican Cochrane Centre–Biomedical 400 Research Institute Sant Pau (IIB Sant Pau)), Christos Christogiannis (University of Ioannina, Department of Primary Education), Claudio Alfonso Rocha Calderón (Iberoamerican Cochrane Centre–Biomedical Research Institute Sant Pau (IIB Sant Pau) and CIBER de Epidemiología y Salud Pública (CIBERESP)), Cordula Wagner (Netherlands Institute for Health Services Research, (NIVEL)), Giorgos Seitidis (University of Ioannina, Department of Primary Education), Jany Rademakers (Netherlands Institute for Health Services Research, (NIVEL)), Katerina-Maria Kontouli (University of Ioannina, Department of Primary Education), Karla Salas (Iberoamerican Cochrane Centre–Biomedical Research Institute Sant Pau (IIB Sant Pau)), Kevin Pacheco-Barrios (Avedis Donabedian Research Institute (FAD)), Kostas Aligiannis (European Patients' Forum), Marieke van der Gaag (Netherlands Institute for Health Services Research, (NIVEL)), Matthijs Michaël Versteegh (Institute for Medical Technology Assessment, Erasmus University Rotterdam, Rotterdam), Montserrat León (Iberoamerican Cochrane Centre–Biomedical Research Institute Sant Pau (IIB Sant Pau)), Nina Adrion (OptiMedis AG), Oliver Groene (OptiMedis AG), Pablo Alonso (Iberoamerican Cochrane Centre–Biomedical Research Institute Sant Pau (IIB Sant Pau) and CIBER de Epidemiología y Salud Pública (CIBERESP)), Rune Poortvliet (Netherlands Institute for Health Services Research, (NIVEL)), Sofia Tsokani (University of Ioannina, Department of Primary Education), Stavros Nikolakopoulos (University of Ioannina, Department of Primary Education), Stella Zevgiti (University of Ioannina, Department of Primary Education), Jessica Zafra (Avedis Donabedian Research Institute (FAD)).

### Patients and public involvement

Patients were a key component of this phase of the COMPAR-EU project. Their interests are represented by the European Patients' Forum (EPF).

## Author Contributions

**Conceptualization:** Monique Heijmans, Ena Niño de Guzmán, Lyudmil Ninov, Dimitris Mavridis, Marta Ballester, Rosa Suñol, Carola Orrego.

**Data curation:** Estela Camus-García, Claudia Valli, Hector Pardo-Hernández, Dimitris Mavridis, Marta Ballester, Rosa Suñol, Carola Orrego.

**Formal analysis:** Estela Camus-García, Monique Heijmans, Claudia Valli, Jessica Beltran, Hector Pardo-Hernández, Dimitris Mavridis, Marta Ballester, Carola Orrego.

**Funding acquisition:** Monique Heijmans, Dimitris Mavridis, Rosa Suñol.

**Investigation:** Estela Camus-García, Ana Isabel González-González, Monique Heijmans, Ena Niño de Guzmán, Claudia Valli, Jessica Beltran, Hector Pardo-Hernández, Lyudmil Ninov, Valentina Strammiello, Kaisa Immonen, Dimitris Mavridis, Marta Ballester, Rosa Suñol, Carola Orrego.

**Methodology:** Estela Camus-García, Monique Heijmans, Ena Niño de Guzmán, Claudia Valli, Jessica Beltran, Hector Pardo-Hernández, Lyudmil Ninov, Valentina Strammiello, Kaisa Immonen, Dimitris Mavridis, Marta Ballester, Rosa Suñol, Carola Orrego.

**Project administration:** Monique Heijmans, Rosa Suñol.

**Resources:** Monique Heijmans, Rosa Suñol, Carola Orrego.

**Supervision:** Monique Heijmans, Ena Niño de Guzmán, Claudia Valli, Jessica Beltran, Dimitris Mavridis, Marta Ballester, Rosa Suñol, Carola Orrego.

**Validation:** Estela Camus-García, Ana Isabel González-González, Monique Heijmans, Ena Niño de Guzmán, Claudia Valli, Dimitris Mavridis, Marta Ballester, Rosa Suñol, Carola Orrego.

**Visualization:** Estela Camus-García, Ana Isabel González-González, Monique Heijmans, Ena Niño de Guzmán, Claudia Valli, Jessica Beltran, Hector Pardo-Hernández, Marta Ballester, Rosa Suñol, Carola Orrego.

**Writing – original draft:** Estela Camus-García, Ana Isabel González-González, Claudia Valli.

**Writing – review & editing:** Estela Camus-García, Ana Isabel González-González, Monique Heijmans, Ena Niño de Guzmán, Claudia Valli, Jessica Beltran, Hector Pardo-Hernández, Lyudmil Ninov, Valentina Strammiello, Kaisa Immonen, Dimitris Mavridis, Marta Ballester, Rosa Suñol, Carola Orrego.

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
