## [Decision Letter · Decision Letter 0]

24 Dec 2020

PONE-D-20-34041

Self-management interventions for adults living with Chronic Obstructive Pulmonary Disease (COPD): The development of a Core Outcome Set for COMPAR-EU project.

PLOS ONE

Dear Dr. González-González,

Thank you for submitting your manuscript to PLOS ONE. After careful consideration, we feel that it has merit but does not fully meet PLOS ONE’s publication criteria as it currently stands. Therefore, we invite you to submit a revised version of the manuscript that addresses the points raised during the review process.

We look forward to receiving your revised manuscript.

Kind regards,

Vijayaprasad Gopichandran

Academic Editor

PLOS ONE

Journal Requirements:

2.) In the Methods section, please provide additional details regarding how researchers and health professionals were recruited for the Delphi surveys, and any eligibility criteria applies to the selection of these participants.

3.) Your ethics statement should only appear in the Methods section of your manuscript.

If your ethics statement is written in any section besides the Methods, please move it to the Methods section and delete it from any other section.

Please ensure that your ethics statement is included in your manuscript, as the ethics statement entered into the online submission form will not be published alongside your manuscript.

4.) Please include a separate caption for each figure in your manuscript.

5.) Please include captions for your Supporting Information files at the end of your manuscript, and update any in-text citations to match accordingly. Please see our Supporting Information guidelines for more information: http://journals.plos.org/plosone/s/supporting-information

Reviewers' comments:

Reviewer's Responses to Questions

**Comments to the Author**

1. Is the manuscript technically sound, and do the data support the conclusions?

Reviewer #1: Yes

Reviewer #2: Partly

2. Has the statistical analysis been performed appropriately and rigorously? 

Reviewer #1: N/A

Reviewer #2: N/A

3. Have the authors made all data underlying the findings in their manuscript fully available?

Reviewer #1: Yes

Reviewer #2: Yes

4. Is the manuscript presented in an intelligible fashion and written in standard English?

Reviewer #1: Yes

Reviewer #2: Yes

5. Review Comments to the Author

Reviewer #1: Upon review it is clear that this manuscript meets the Plos One requirements. There are, however, some minor points to consider or need a bit of detail to explain, please see comments in attached document.

Reviewer #2: Self-management interventions for adults living with Chronic Obstructive Pulmonary Disease (COPD): The development of a Core Outcome Set for COMPAR-EU project.

This project is part of the COMPAR-EU and supported by EU Horizon 2020 research and innovation programme (grant agreement no. 754936).

COMPAR-EU is a multimethod, interdisciplinary project that aims to contribute bridging the gap between current knowledge and practice of self-management interventions (SMI).

It involves patients to establish priorities and preferences.

Self-management is defined as `what individuals, families and communities do with the intention to promote, maintain, or restore health and to cope with illness and disability with or without the support of health professionals. It includes but is not limited to self-prevention, self-diagnosis, self-medication and self-management of illness and disability.´

COMPAR-EU aims to identify, compare, and rank the most effective and cost-effective self-management interventions (including preventive and management domains) in Europe for adults suffering from one of the four high-priority chronic diseases: type 2 diabetes, obesity, chronic obstructive pulmonary disease (COPD), and heart failure.

COMPAR-EU publications

1. Perry-Duxbury M, Asaria M, Lomas J, Van Baal P. Cured Today, Ill Tomorrow: A Method for Including Future Unrelated Medical Costs in Economic Evaluation in England and Wales Value Health 2020 Aug;23(8):1027-1033.

2. Ballester M, Orrego C, Heijmans M, et al. Comparing the effectiveness and cost- effectiveness of self- management interventions in four high- priority chronic conditions in Europe (COMPAR- EU): a research protocol. BMJ Open 2020;10:e034680. doi:10.1136/bmjopen-2019-034680

SUMMARY OF THE STUDY/MANUSCRIPT SUBMITTED

This manuscript report on the COMPAR-EU study specific objective 2 and for the chronic disease COPD: To identify and prioritise self-management intervention (SMI) outcomes from patients´ perspectives using systematic literature review and Delphi survey with patient and career representatives to ensure that the included outcomes truly reflect patients´ priorities and preferences.

SMI for COPD is challenging due to heterogeneity of interventions, lack of clear definitions of self-management components, and variability in the outcomes reported. Moreover, systematic reviews on SMI effectiveness have found insufficient data for some outcomes, which may be suggestive of selective reporting.

By reaching consensus of a standardized set of outcomes that should be minimally measured and reported in future COPD clinical trials, we will ensure the comparativeness of results and synthesis of the evidence across studies. This outcome set should be relevant for all stakeholders, but especially for patients, as they are the ones primarily responsible for the daily management of their disease.

In this study they propose a systematic approach to develop a Core Outcome Set [COS] for measuring the effectiveness of SMIs interventions in COPD from the perspective of both patients and health care professionals.

Methods - They undertook an innovative approach consisting of four interlinked stages: i) Development of an initial catalogue of outcomes from previous EU-funded projects and/or published studies, ii) Scoping review of reviews on patients and caregivers’ perspectives to identify outcomes of interest, iii) Two-round Delphi online survey with patients and patient representatives to rate the importance of outcomes, and iv) Face- to-face consensus workshop with patients, patient representatives, health professionals and researches to develop the COS.

Results- From an initial list of 79 potential outcomes, 16 were included in the COS plus one supplementary outcome relevant to all participants. These were related to patient and caregiver knowledge/competence, self-efficacy, patient activation, self-monitoring, adherence, smoking cessation, COPD symptoms, physical activity, sleep quality, caregiver quality of life, activities of daily living, coping with the disease, participation and decision-making, emergency room visits/admissions and cost effectiveness.

See Fig 2 and table 3

Conclusion- The development of the COPD COS for the evaluation of SMIs will increase consistency in the measurement and reporting of outcomes across trials. It will also contribute to more personalized health care and more informed health decisions in clinical practice as patients’ preferences regarding COPD outcomes are more systematically included.

REVIEW AND COMMENTS

This study and manuscript are well done and written.

There is no gold standard method for the development of a COS.

The process for selecting a core outcome set requires a rigorous methodology in which outcomes are first pooled from all possible sources and then subsequently prioritized during a consensus process.

This research is done in accordance to the Core Outcome Measures for Effectiveness Trials (COMET) and the Core Outcome Set-STAndards for Development (COS-STAD) guidelines. This guideline is well known and has been the topic of many publications.

The study protocol has already been published.

It is clear from reading the manuscript that the COS-STAD recommendations were developed to address the first stage of development, namely gaining agreement on what should be measured.

However, it is not clear and this needs to be specified what should be measured in a particular research or practice setting, with subsequent work needed to determine how each outcome should be defined or measured, lack of selection of outcome instruments and the follow-up timing.

For each of the 4 interlinked stages described and used on this study, I would have the following comments and suggestions:

Stage 1 and 2: Development and initial catalogue of outcomes (Stage 1), and scoping review on perspectives of patients and caregivers (Stage 2)

The following questions reflect important limitation with Stage 1 and 2 of the process:

i) Why limiting the review to EU-funded projects and/or published studies?

ii) Why having the main focus of the review using drug trials and not also non drug/behavioral intervention trials in COPD as well?

iii) Why not having qualitative research included as well? As part of the early process, focus groups could be used to elicit participants’ expectations regarding SMI COPD. Relying only on the existing literature and scoping review may miss some of the important outcome patients or care givers may expect. The study has a strong design, which is based on a thorough methodological review but limited qualitative research to develop a longlist of clinically relevant outcomes.

These limitations should be at minimum discuss as potential limitations and for subsequent work.

Stage 3: Delphi survey:

You invited adults diagnosed with COPD and patient’ representatives.

Why not having other representatives such as HCP (e.g. doctors, nurses, physiotherapists), researchers (health professionals who care for patients but are also involved in designing research studies). Experts in COPD self-management, especially those in Europe. This should at minimum be discussed.

It would have been of value to have other representatives and to contrast with patient perspectives. These two perspectives could complete each other.

How did you ensure the representativeness of the patients with COPD severity (GOLD 1 to 4, high vs low symptom burden and high vs low risk of exacerbations), male and female, high and low socioeconomic status, few vs many co-morbidities.

Sample size is usually not calculated for this type of studies but how did you ensure that saturation was achieved and potential representativeness differences were captured.

Consensus meeting:

For the consensus, you invited COPD patients and patients’ representatives who participated in the two Delphi rounds, health professionals and researchers.

Why and how did you come up with a maximum of 15 outcomes.

Relevant arguments for or against the inclusion of an outcome and the vote counts should be noted, and reported or given access.

Results

The study characteristics reflect the limitation of selecting primarily outcome in COPD from drug trials.

The outcome classification in Table 2 should also inform the reader on the frequency of the different outcomes reported.

Table 3 is very comprehensive using a theme classification.

Discussion

The discussion is very limited, in particular very little is provided in the context of the existing literature. It may be that very little has been done on this topic in COPD. At least, the discussion should provide the similarities and differences. Then what this study is adding and/or justification of why it may have differences.

Furthermore, the discussion doesn’t give the impression that the authors are aware of what the study was not able to cover. I have pointed out many of them and those should be addressed in the discussion.

It is to me an overstatement “We are confident that the COPD COS reflects the preferences of all key stakeholders and that it might be applicable with context adaptation to wide range of settings across Europe and the world. ” The study methodology used in this study allows to say the first part of the statement, eg., it reflects patient preferences. However, the design of the study doesn’t allow to conclude that it might be applicable in a wide range of settings across Europe and the world.

This study is novel and a good first stage but future work should be done to validate the COPD COS before the authors can claim the universality.

6. PLOS authors have the option to publish the peer review history of their article (what does this mean?). If published, this will include your full peer review and any attached files.

Reviewer #1: **Yes: **Maarten Voorhaar

Reviewer #2: No

---

## [Author Response · Author response to Decision Letter 0]

7 Feb 2021

Revisions (R) made according to Plos One reviewer's report (Reviewer #1) by queries (Q):

Reviewer #1: Upon review it is clear that this manuscript meets the Plos One requirements. There are, however, some minor points to consider or need a bit of detail to explain, please see comments in attached document.

We would like to thank Reviewer 1 for the positive comments and the feedback provided. Below we address each comment and suggestion.

Q1 - Possible typo: Researches or researchers?

R1 – We meant researchers. Apologies for the mishap.

• Abstract (Lines 103-105): “Face-to-face consensus workshop with patients, patient representatives, health professionals and researchers to develop the COS”.

Q2 - replace '...one of them being...' with 'including'

R2 – We thank Reviewer #1 for the suggestion and modified the section accordingly as follows:

• Introduction (lines 151-153): “This study is part of COMPAR-EU, an EU-funded project designed to bridge the gap between current knowledge and practice on SMIs in four chronic conditions one of them being including COPD”

Q3 - What is the reasoning behind inclusion of EU funded projects only? Does this mean non-EU projects were excluded? Please clarify

R3 – We did not exclude non-EU projects. Instead, we built upon those EU funded projects that consisted of overviews of systematic reviews evaluating the effectiveness of SMI in patients with chronic diseases, including COPD. We did not exclude any of the systematic reviews in terms of the country of development. We have rephrased the text as follows:

• Material and methods – Stage 1 – Development of an initial catalogue of outcomes – Data sources and searches (lines 164-174): “We developed an initial catalogue of outcomes from a literature review of previous EU-funded projects that included two overviews of systematic reviews focusing on evaluating the effectiveness of SMIs for chronic diseases: i) PRO-STEP (Promoting Self-Management for Chronic Diseases in the EU) (17) and ii) EMPATHiE (Empowering Patients in the Management of Chronic Diseases) (18). Both reviews (17,18) were performed by the research team and were considered as the starting data source to build the initial list of outcomes. We additionally searched for COPD COS in relevant organizations databases such as COMET (10) and ICHOM [International Consortium for Outcomes Health Measurement] (19),.”

Q4 - Please indicate if the referee was or was not one of the authors

R4 – We appreciate the suggestion and tried to clarify what we meant as follows:

• Material and methods – study selection (lines 186-188): “We screened title and abstracts and assessed eligible full-text articles independently. In case of disagreement, reviewers reached consensus or consulted with a third reviewer from the review teamreferee.”

Q5 - to identify

R5 – We thank for the identification of the typo and corrected accordingly:

• Material and methods – Stage 2 (Lines 212-214): “We conducted a scoping review of reviews (21) to identify and describe key concepts related to outcomes by exploring patients’ and caregivers’ preferences and experiences when coping with COPD and its self-management.

Q6 - Please specify or give examples of these terms

R6 – We have added the terms as shown below:

• Material and methods – Stage 1 - Data sources and searches (Lines 176-180): “The syntax used for the additional literature review in PubMed was the following: (pulmonary disease, chronic obstructive "[MeSH Terms] AND "patient preference"[MeSH Terms]) AND "outcome assessment (health care)"[MeSH Terms]; "pulmonary disease, chronic obstructive"[MeSH Terms] AND "core outcome set"[All Fields].”

• Material and methods – Stage 3 - Data sources and searches (Lines 218-220): “We included the following terms for identifying patients’ perspectives: patient perception, experience, perspective, understanding, preferences and health utilities.”

Q7 - Please describe to which extent these patients are representing average COPD patients? (i.e. age, education, personal beliefs/ empowerment)

R7 – We added a more detailed explanation:

• Material and methods - Stage 3 - Study population and eligibility criteria (lines 241-254): “We included adults diagnosed with COPD and patient representatives who were able to understand and speak English and provided informed consent to participate through the web platform hosting the Delphi rounds. We made efforts to recruit patients considering age, gender geographical location and education. However, the patients who participated in this study may have been more knowledgeable, motivated and aware of treatment options and legislation than other COPD patients. In the other hand, they may have been more motivated to engage in research and advocacy activities. They may have also been more aware of the needs of other COPD patients and during the discussion it was evident that they wanted to represent the views of COPD patients as a whole and not just their own. As an example, they mentioned that while they were aware of strategies to avoid exacerbations, other patients may be less knowledgeable.” 

Q8 - Were participants able and/or allowed to communicate among each other? Please specify

R8 – Yes. During the discussions, participants were allowed to deliberate.

• Material and methods - Stage 3 - Delphi survey (lines 270-273): “During the second round, participants were able to see ratings (average score) from the first round and thus, adjust, confirm or rethink their answers. They were also allowed to deliberate. This process enabled participants to rate the most relevant SMIs outcomes for COPD according to their perspective.”

Q9 - Please describe the term 'snowballing'

R9 – By snowballing we meant searching relevant documents and then: 1) looking at suggestions of similar studies in the search databases; 2) looking at the references of eligible studies; and 3) re-running searches using terms from eligible studies. 

• Results – Stage 1 – Study selection (lines 326-329): “The additional search in COMET (10), ICHOM (19) and through snowballing, which included i) looking at suggestions of similar studies in the search databases, ii) looking at the references of eligible studies, and iii) re-running searches using terms from eligible studies, yielded 23 articles.”

Q10 - Please describe in the method section why 70% was considered a cut off for high agreement

R10 – We chose 70% based on guidelines and previous publications. We have added the information and references as follows: 

• Material and methods – Stage 3 – Delphi survey – Data synthesis and analysis (lines 275-281): “All outcomes were categorized into three groups based on the level of agreement of ratings from the two-round Delphi online surveys as follows (Table 1): i) Group 1 or “high consensus and high importance outcomes”, ii) Group 2 or “low consensus and mixed importance outcomes” and iii) Group 3 or “high consensus of moderate and low importance outcomes”. We used 70% as a cut off for high agreement based on GRADE recommendations, COMET guidelines and previous papers reporting patient-centred core outcome sets that also used these thresholds (20,25,26).”

Q11 - Where does former refer to?

R11- After reviewing this section, we realized that it was best to leave this sentence out, as follows:

• Results – Stage 4 - Consensus workshop and final COPD core outcome set (lines 378-381). “The former, coping with the disease, including depression and anxiety, was included in the final version of the COS (as part of the quality of life category and psychological functioning subcategory.

Q12 - Missing bracket.

R12 – We finally deleted the sentence (see Q11).

Revisions (R) made according to Plos One reviewer's report (Reviewer #2) by queries (Q):

Q13 - This study and manuscript are well done and written. There is no gold standard method for the development of a COS. The process for selecting a core outcome set requires a rigorous methodology in which outcomes are first pooled from all possible sources and then subsequently prioritized during a consensus process. This research is done in accordance to the Core Outcome Measures for Effectiveness Trials (COMET) and the Core Outcome Set-STAndards for Development (COS-STAD) guidelines. This guideline is well known and has been the topic of many publications. The study protocol has already been published.

R13 - We thank Reviewer #2 for the thorough assessment of our manuscript and for his positive feedback.

Q14 - It is clear from reading the manuscript that the COS-STAD recommendations were developed to address the first stage of development, namely gaining agreement on what should be measured. However, it is not clear and this needs to be specified what should be measured in a particular research or practice setting, with subsequent work needed to determine how each outcome should be defined or measured, lack of selection of outcome instruments and the follow-up timing.

R14 - The reviewer raises very important issues related to how to measure the identified outcomes and how to collect relevant information. We feel these issues are beyond the scope of this paper and should be the focus of future research. We have updated the Discussion/Implications for practice and research section as follows:

• Discussion – Implications for practice and research (lines 472-477): “Further work is needed to identify and provide guidance on the most appropriate measures for each outcome, on the right instruments or approaches to measure these outcomes, and on the length of follow up. Moreover, it will be important to identify strategies for fostering the collection of this information, the role of the different providers, and the settings where these outcomes can be assessed.” 

Q15 - For each of the 4 interlinked stages described and used on this study, I would have the following comments and suggestions:

Stage 1 and 2: Development and initial catalogue of outcomes (Stage 1), and scoping review on perspectives of patients and caregivers (Stage 2)

The following questions reflect important limitation with Stage 1 and 2 of the process:

i) Why limiting the review to EU-funded projects and/or published studies?

ii) Why having the main focus of the review using drug trials and not also non drug/behavioral intervention trials in COPD as well?

iii) Why not having qualitative research included as well? As part of the early process, focus groups could be used to elicit participants’ expectations regarding SMI COPD. Relying only on the existing literature and scoping review may miss some of the important outcome patients or care givers may expect. The study has a strong design, which is based on a thorough methodological review but limited qualitative research to develop a longlist of clinically relevant outcomes.

These limitations should be at minimum discuss as potential limitations and for subsequent work.

R15 – We answer to the suggestions bellow:

i) This question has been already answered in Q3. We did not limit the review to EU-funded projects neither to published trials. What we did was to identify the list of outcomes from two overviews of systematic reviews evaluating the effectiveness of self-management interventions. These overviews were funded by the European Commission. We have clarified and improve the wording of this section.

ii) Our focus was not reviewing drug trials, just the opposite. Our focus was identifying outcomes from systematic reviews evaluating the effectiveness of self-management interventions. We additionally look for sets of outcomes that including patients for their selection. We considered that taking into account these patient-centered outcomes was a more comprehensive way to build the list to be showed to the patients, even if the focus was on drug-trial. We only found three studies corresponding to these characteristics.

iii) We agree with the reviewer on the importance of qualitative research for eliciting relevant outcomes for patients. We have used them in different stages of our research:

a. In stage 1: For the additional search of core outcome sets, we did not exclude the papers by research methods. In fact, the studies included qualitative methods or mixed.

b. In stage 2: the methods included review of quantitative, qualitative or mixed studies. In fact, taking into account the results from this stage, 59%of the were qualitative evidence synthesis.

Finally, the participatory procedures were based on qualitative research, where the patients had the opportunity to add outcomes from the very beginning. 

Q16 - Stage 3: Delphi survey: You invited adults diagnosed with COPD and patient representatives. Why not having other representatives such as HCP (e.g. doctors, nurses, physiotherapists), researchers (health professionals who care for patients but are also involved in designing research studies). Experts in COPD self-management, especially those in Europe. This should at minimum be discussed.

It would have been of value to have other representatives and to contrast with patient perspectives. These two perspectives could complete each other.

How did you ensure the representativeness of the patients with COPD severity (GOLD 1 to 4, high vs low symptom burden and high vs low risk of exacerbations), male and female, high and low socioeconomic status, few vs many co-morbidities.

Sample size is usually not calculated for this type of studies but how did you ensure that saturation was achieved and potential representativeness differences were captured.

R16 – Our main objective was to identify a set of outcomes that were relevant for patients. 

Q17 - Consensus meeting: For the consensus, you invited COPD patients and patient representatives who participated in the two Delphi rounds, health professionals and researchers.

Why and how did you come up with a maximum of 15 outcomes.

Relevant arguments for or against the inclusion of an outcome and the vote counts should be noted, and reported or given access.

R17 - The development of a core outcome set has the potential to reduce heterogeneity between trials and lead to research that is more likely to have measured relevant outcomes. Identifying outcomes of particular relevance when evaluating the effects of self-management interventions required prioritizing those most important. This process also makes more efficient the comparative effectiveness research among trials. We decided 15 outcomes as a threshold to facilitate our stakeholders to discern on those most relevant. However, we left open the option of including supplementary outcomes to not “force” them to exclude outcomes that were considered important but obtaining slightly lower scoring. Furthermore, based on what it had been published and the purpose of COMPAR-EU project, we decided that a list of 15 outcomes was a good number of outcomes, allowing covering different categories (empowerment, clinical outcome, quality of life, healthcare use, etc.) and also feasible for performing comparison among trials. The number of core outcome sets is undetermined by guidelines and recommendations, but in the light of our results, and having only one supplementary outcome, we can assume that it was the right decision. 

Q18 – Results: The study characteristics reflect the limitation of selecting primarily outcome in COPD from drug trials. The outcome classification in Table 2 should also inform the reader on the frequency of the different outcomes reported.

Table 3 is very comprehensive using a theme classification.

R18 - Most of the studies included in our initial review emerged from trials focused on self-management interventions. All the systematic reviews included in the overviews were based on randomized control trials evaluating the effectiveness of self-management interventions. The list of outcomes presented in table 2 mostly emerged from self-management interventions. However, we also considered publications on core outcome sets specific for COPD (even if they were not addressed to SMIs). We made this decision with the purpose of having a comprehensive list of outcomes and ensuring that previous prioritization processes including patients were considered. We have rephrased the text and make clear characteristics of all the included studies. Table 2 was built on an iterative process, including refinement, merging of similar outcomes, and combining the information from stage 1 and 2. As we based our review on secondary synthesis of the literature, including the frequency of the studies containing the outcomes may lead confusion on their real use in clinical trials.

Q19 – Discussion: The discussion is very limited, in particular very little is provided in the context of the existing literature. It may be that very little has been done on this topic in COPD. At least, the discussion should provide the similarities and differences. Then what this study is adding and/or justification of why it may have differences.

Furthermore, the discussion doesn’t give the impression that the authors are aware of what the study was not able to cover. I have pointed out many of them and those should be addressed in the discussion.

It is to me an overstatement “We are confident that the COPD COS reflects the preferences of all key stakeholders and that it might be applicable with context adaptation to wide range of settings across Europe and the world. ” The study methodology used in this study allows to say the first part of the statement, e.g., it reflects patient preferences. However, the design of the study doesn’t allow to conclude that it might be applicable in a wide range of settings across Europe and the world.

This study is novel and a good first stage but future work should be done to validate the COPD COS before the authors can claim the universality.

R19 – Thanks for the point, the little discussion is due to space limitation, we focused on the design, the development and explanation of the COS. As the reviewers says, very little has been done on this topic in COPD.

 

Journal Requirements: When submitting your revision, we need you to address these additional requirements.

Q20 - Please ensure that your manuscript meets PLOS ONE's style requirements, including those for file naming. The PLOS ONE style templates can be found at

R20 – Many thanks, we will do.

Q21 - In the Methods section, please provide additional details regarding how researchers and health professionals were recruited for the Delphi surveys, and any eligibility criteria applies to the selection of these participants.

R21 – Thanks for raising this point out. 

Patients for the Delphi on-line surveys were identified by the European Patients Forum, EU wide membership network of 70+ patient organization to seek an optimal balance between gender, graphic area, and English skills. And from those, we recruited participants along EU countries who were able to resend the informed consent, to understand and answering the online Delphi questionnaires and traveling to take part in consensus workshop. 

Researchers and Health professionals for the consensus meeting were selected by a purposive sample of a heterogeneous group of health professionals representing relevant specialties on the care of patients with COPD (general practitioners, specialists, nurses…) and researchers that came from the seven collaborating partners-teams, those who knew the process well, and who would focus on the ultimate objective of facilitating dialogue between patients, representatives and professionals during the consensus workshop. We specified in the text as follows:

• Stage 4 – Consensus workshop and final COPD COS (lines 289-296): “Researchers and health professionals were selected from a purposive sample of a heterogeneous group of health professionals representing relevant specialties on the care of patients with COPD (general practitioners, specialists, nurses…) and researchers that came from seven collaborating partner-teams, who knew the process well and could participate on the ultimate objective of facilitating dialogue between patients, patient representatives and health professionals during the consensus workshop.”

Q22 - Your ethics statement should only appear in the Methods section of your manuscript. If your ethics statement is written in any section besides the Methods, please move it to the Methods section and delete it from any other section.

Please ensure that your ethics statement is included in your manuscript, as the ethics statement entered into the online submission form will not be published alongside your manuscript.

R22 – We modified accordingly.

Q23 - Please include a separate caption for each figure in your manuscript.

R23 – We will do.

Q24 - Please include captions for your Supporting Information files at the end of your manuscript, and update any in-text citations to match accordingly. Please see our Supporting Information guidelines for more information: http://journals.plos.org/plosone/s/supporting-information

R24 – We will do.

---

## [Editor Report · Decision Letter 1]

9 Feb 2021

Self-management interventions for adults living with Chronic Obstructive Pulmonary Disease (COPD): The development of a Core Outcome Set for COMPAR-EU project.

PONE-D-20-34041R1

Dear Dr. González-González,

We’re pleased to inform you that your manuscript has been judged scientifically suitable for publication and will be formally accepted for publication once it meets all outstanding technical requirements.

Kind regards,

Vijayaprasad Gopichandran

Academic Editor

PLOS ONE
---

## [Editor Report · Acceptance letter]

15 Feb 2021

PONE-D-20-34041R1 

Self-management interventions for adults living with Chronic Obstructive Pulmonary Disease (COPD): The development of a Core Outcome Set for COMPAR-EU project. 

Dear Dr. González-González:

I'm pleased to inform you that your manuscript has been deemed suitable for publication in PLOS ONE. Congratulations! Your manuscript is now with our production department. 

Kind regards, 

on behalf of

Dr. Vijayaprasad Gopichandran 

Academic Editor

PLOS ONE